# Fast Response to Superspreading: Uncertainty and Complexity in the Context of COVID-19

**DOI:** 10.3390/ijerph17217884

**Published:** 2020-10-27

**Authors:** Lukas Zenk, Gerald Steiner, Miguel Pina e Cunha, Manfred D. Laubichler, Martin Bertau, Martin J. Kainz, Carlo Jäger, Eva S. Schernhammer

**Affiliations:** 1Department of Knowledge and Communication Management, Faculty of Business and Globalization, Danube University Krems, 3500 Krems an der Donau, Austria; gerald.steiner@donau-uni.ac.at; 2Complexity Science Hub Vienna, 1090 Vienna, Austria; manfred.laubichler@asu.edu (M.D.L.); jaeger@globalclimateforum.org (C.J.); 3Nova School of Business and Economics, Universidade Nova de Lisboa, 2775-405 Carcavelos, Portugal; miguel.cunha@novasbe.pt; 4School of Complex Adaptive Systems Tempe, Arizona State University, Tempe, AZ 85287-2701, USA; 5Santa Fe Institute, Santa Fe, NM 87501, USA; 6Global Climate Forum, 10178 Berlin, Germany; 7Institute of Chemical Technology, Freiberg University of Mining and Technology, 09599 Freiberg, Germany; martin.bertau@chemie.tu-freiberg.de; 8WasserCluster Lunz-Inter-University Center for Aquatic Ecosystem Research, 3293 Lunz am See, Austria; martin.kainz@donau-uni.ac.at; 9Academy of Disaster Reduction and Emergency Management, Beijing Normal University, Beijing 100875, China; 10Department of Epidemiology, Center for Public Health, Medical University of Vienna, 1090 Vienna, Austria; 11Channing Division of Network Medicine, Harvard Medical School, Boston, MA 02115, USA

**Keywords:** complex systems, COVID-19, superspreading, networks, fast response, improvisation, interdisciplinary perspectives, transdisciplinarity, SARS-CoV-2, pandemic

## Abstract

Although the first coronavirus disease 2019 (COVID-19) wave has peaked with the second wave underway, the world is still struggling to manage potential systemic risks and unpredictability of the pandemic. A particular challenge is the “superspreading” of the virus, which starts abruptly, is difficult to predict, and can quickly escalate into medical and socio-economic emergencies that contribute to long-lasting crises challenging our current ways of life. In these uncertain times, organizations and societies worldwide are faced with the need to develop appropriate strategies and intervention portfolios that require fast understanding of the complex interdependencies in our world and rapid, flexible action to contain the spread of the virus as quickly as possible, thus preventing further disastrous consequences of the pandemic. We integrate perspectives from systems sciences, epidemiology, biology, social networks, and organizational research in the context of the superspreading phenomenon to understand the complex system of COVID-19 pandemic and develop suggestions for interventions aimed at rapid responses.

## 1. Introduction

The new coronavirus disease 2019 (COVID-19) has revealed how interconnected our world has become. The actual degree of complexity becomes apparent if one extends the perspective beyond the societal impact of the human disease COVID-19 to coupled systems of humans and nature. Here, we need to consider the emergence of zoonotic viruses such as the severe acute respiratory syndrome coronavirus 2 (SARS-CoV-2) and the effects of different time scales. In the near term, higher connectivity in natural ecosystems can increase the stability of these systems; however, in the long term, effective evolvability often depends on a more modular organization. Natural systems have to navigate these conflicting demands. For example, in natural ecosystems, higher connectivity between resources and consumers (trophic dependence) favors the stability of these systems [1]. However, highly connected systems may also be weakened when essential resources are removed. This can cause a decline of established predator–prey relationships to the point that the survival success of the consumers may become very low [2] and the whole established system could eventually collapse, allowing the development of and new systems. A fundamental anthropogenic problem that has more recently emerged is that the delicate balance between stability and evolvability of these systems has been disturbed by human action, which may increase the probabilities of zoonotic viruses. When considering social systems, more interconnected networks also have positive and negative impacts, visible at different scales from the local to the global level. On the one hand, strongly interconnected social systems may facilitate more efficient exchange and use of essential resources and social capital. However, as evidenced by pandemic threats such as COVID-19, denser social networks can also act as catalysts for crises, turmoil, and destruction.

One hypothesis regarding the emergence of viral outbreaks is that human-induced reductions in the habitat condition of host species, such as bats, may make further virus outbreaks more likely [3,4,5]. The high interconnectedness of social and economic systems also enables viruses to spread particularly quickly and efficiently. In such dynamic environments, system resilience is of crucial importance. It is characterized by the adaptability of coupled human–nature systems to disturbances from within or in their environment. These adaptive capacities have to be strengthened by specific interventions and strategies in a proactive or reactive manner before, during, and after such threats as the SARS-CoV-2. A sufficient and rapid understanding of the behavior of complex systems, including relevant factors and relations, embedded in a highly uncertain environment is an important determinant for such sustainable system interventions [6,7], in terms of both pandemic threat preparedness and the ability to respond rapidly by organizations, countries, and the international community as a whole in the event of a pandemic. The modeling and simulation of epidemics allows investigating possible future scenarios. However, a major challenge for modeling these highly interconnected systems and to assess the effects of possible interventions is the current lack of detailed data on all relevant dimensions [8]. Continuous data collection and the application of approaches such as info-metrics [9] will successively improve the basis for decision-making in the face of uncertainty.

To determine appropriate measures in response to a pandemic threat, the collected data and the level of knowledge about the complexity of the involved systems at a given point in time is crucial [10]. In the case of COVID-19, a few infected persons who spread the virus worldwide in a very short time were apparently the first trigger of the pandemic. In addition, there are indications that certain individuals and events, the so-called superspreaders, will particularly intensify the spread, thus further increasing uncertainty and challenging measures to contain the situation. What we face is the challenge of developing a rapid understanding of complex situations in face of rapidly escalating problems in order to develop appropriate responses, which is particularly difficult when closely coupled systems interact strongly. Science in its ideal, for example, is a system that continuously develops quality knowledge through transparent methods, including peer-reviewed processes that take a considerable amount of time. In contrast, decision-makers must act and intervene immediately in the event of outbreaks, basing their decisions on incomplete knowledge and few known incidents [11]. We argue that two key dimensions largely determine the potential success of short- and long-term interventions to address such uncertainties and threats: (1) integrated knowledge of complex systems based on experience from different sectors; (2) government systems and streamlined organizational structures that support long-term preparedness and enable rapid response.

Superspreaders represent a specific form of uncertainty with an extreme leverage effect, which makes them a crucial starting point for immediate action in the acute phase of a viral threat such as SARS-CoV-2. Therefore, understanding the propagation mechanisms of superspreaders and designing organizational frameworks that allow controlling their impact is an essential component of a successful intervention portfolio. With this in mind, the paper is organized into two major parts: (1) what we currently understand about superspreading of COVID-19 and (2) how to deal with uncertain situations due to superspreading events in the sense of “managing the unexpected” [12], where preparation for improvisation becomes indispensable.

## 2. Understanding Superspreading

### 2.1. Epidemiological Parameters of a Spread

From an epidemiological perspective, a well-known measure of the transmission dynamics of a particular virus is the basic reproduction number (R0), which gives an impression of the speed at which a disease spreads [13]. With R0 = n, one person on average infects n other persons; R0 > 1 indicates an increase of the spread as the number of cases is likely to increase [14]. In the early stage of the outbreak, several studies estimated R0 for COVID-19 to be considerably higher than 1 [15,16,17]. However, this indicator may be misleading, as it is an averaged number that does not accurately represent the actual spread. In fact, recent evidence points to another important aspect complementing the average transmission dynamics R0: the secondary attack rate (SAR). The SAR captures the likelihood of an infection occurring within a close setting such as among household members or in spaces with close contacts, as for example during the now infamous après-ski parties in Ischgl [18]. Studies have shown that the SAR can be up to 35% (95% confidence interval, 27–44%) [19] and even higher, depending on the setting [20]. This is crucial, considering that, if each of a group of 10 infects one other person, R0 is 1; however, without considering SAR, R0 would also remain 1 in a highly different scenario, that is, if one of a group of 10 infects 10 other people, whereas the remaining 9 of the group do not infect anyone. Depending on these scenarios, which both have an R0 = 1, the spread would be remarkably different.

To take this variance into account, the dispersion parameter (K) is often used as a key parameter in studying outbreak dynamics. K is helpful in estimating the individual difference among infectious people transmitting a pathogen (i.e., whether some individuals infect few, and others infect large numbers of contacts, or whether each infected person infects roughly the same number of other individuals). A K number above 1 indicates low variation in spreading patterns, i.e., infected persons infect about the same number of others. A K number below 1 indicates high variation, i.e., certain persons infect a considerably higher number of persons than others do. A low K value implies that higher transmission comes from a small number of infectious persons, the “superspreaders”.

According to the World Health Organization, a superspreader incident is an individual person or event that transmits an infection to a number of individuals larger than what would be considered usual [21]. Even before the COVID-19 pandemic-related lockdowns, the K of this disease was estimated between 0.1 and 0.5 [22,23]. This would mean that about 10–20% of COVID-19-infected persons had infected 80% of those infected so far, due to heterogeneities in the transmission of infectious agents [24]. A low K brings both good and bad news. On the one hand, it makes it easier to track local outbreaks: if it is possible to identify superspreaders, their contacts can be informed and isolated, and a further outbreak prevented. However, on the downside, it also means increased uncertainty in the sense that even after a pandemic has been largely contained, infection rates can quickly rise again (triggering, in the worst case, a second wave), and uncertain environments persist. Individual superspreaders might, for example, infect a large number of people in a short period of time during a larger event, resulting in super-spreading events.

### 2.2. Characteristics and Networks of Superspreaders

Based on our current knowledge, superspreading is emerging as an important aspect of COVID-19. While still in the midst of a smoldering pandemic and waiting for therapeutic advances or vaccination, it appears paramount to work towards identifying common features of superspreading incidents. Further, to gain a deeper understanding of superspreaders, their individual characteristics as well as their embeddedness in social networks have to be better understood and taken into account.

Several behavioral and biological features have preliminarily been described to characterize a COVID-19 superspreader, though the heterogeneity of spreading must be discussed cautiously, and much remains to be learned [25]. For example, excessive mouthwatering (e.g., increased saliva production due to certain medication), speaking while eating, speaking loudly or shouting (e.g., because of a noisy work environment, hearing impairments, choir singing [26], etc.) and, more generally, wet pronunciation—all leading to spittle spray during speech (the so called “corona cloud”) and hence higher droplet load—appear among some of the more prevalent features at the core of previous superspreader events. Further, a higher virus load [27] and, hence, potentially higher viral shedding [28], perhaps triggered by a weaker immune system, as well as more frequent coughing [29] and cough-generated aerosols [30], have also been described as biological features of some superspreaders.

A person’s social behavior may add to these personal biological and behavioral characteristics: physical contacts and therefore the social network in which a person is embedded are equally essential for a better understanding of superspreading [14]. For example, persons with overly wet pronunciation may become a superspreader only if they are in unprotected contact with a large number of other persons but would not if they are at home or protected. In fact, depending on a person’s social embeddedness including her or his circle of friends and family, interaction with colleagues and clients, traveling activities or professional characteristics, social networks may vary greatly. Understanding these patterns of human interactions and the consequences for transmission dynamics is essential to understand the underlying social process of superspreading incidents. Similar to the importance of K when considering R0 and SAR, the amount of physical contacts of people may not be equally distributed but rather be represented as scale-free networks. In such networks, the distribution follows a power law with a fat tail, which means that few people have many contacts and many people have few contacts [31].

There are several reasons why certain people have a higher degree centrality (people with a high number of contacts) [32]. For example, preferential attachment describes the social process of people with already many contacts getting even more contacts (e.g., politicians, celebrities), i.e., “the rich getting richer” [33,34]. Homophily theories explain that similar people are more likely to network with each other (e.g., young skiers in Ischgl), i.e., “birds of a feather flock together” [35]. Both these human interactions may define a specific group of people with a higher eigenvector centrality, such as people with many physical contacts (e.g., healthcare professionals) who are in contact with other people with many physical contacts (e.g., other healthcare professionals [14]). The tendency of triadic closure (people connecting with mutual acquaintances) further reinforces this development of contagion clusters [36]. As a consequence, in accordance with the dispersion parameter K for COVID-19 discussed earlier and contrary to the idea of equally distributed contacts and a basic reproduction factor as an average value, dense clusters of people with a different amount of social contacts are a more appropriate picture of social networks between humans than an equally distributed network [37,38,39].

These dense clusters are not isolated social islands but are interlinked with other clusters. The more contacts a person has, the more likely these contacts connect otherwise isolated clusters. People with such a high betweenness centrality (people who are in-between others), represent a social position that connects different clusters and thus enables information or viruses to spread faster due to the reduction of the average path length between people. Half a century ago, Milgram [40] hypothesized such a “small world”, the phenomenon that we are only a few handshakes away from each other. Watts and Strogatz [41] explained the small world phenomenon or the so-called “six steps of separation” mathematically as systems that are highly clustered like structured networks but also have short average path lengths like random networks. These small-world networks describe dense clusters of homophile individuals which have an unevenly distributed amount of contacts, as well as interconnections to other clusters that provide a fast spread. These characteristics of networks may have a socio-economic advantage in human systems (in the sense of global innovation, knowledge generation, or global goods transfer) and an evolutionary advantage in natural systems (in the sense of robustness or interconnected food webs [33]). However, in the event of a virus outbreak, these human patterns can lead to sudden local and global increases in viral diffusion, as viruses do not spread evenly and continuously over time but can grow exponentially in a very short time [42].

Although various empirical data were collected over the last decades to analyze social networks, we still need more empirical data to analyze the actual contact networks and events of the COVID-19 pandemic [43,44,45]. This specific “corona network” may have different characteristics than other social networks, such as information exchange networks, and may also be different from other virus networks. Human immunodeficiency virus (HIV) or Ebola networks, for example, are rather sparse due to the different viral transmission mechanisms [46] compared to the much denser network of the corona network, which can have a considerable impact on the spread. Zhangbo [47] investigated the patients’ dynamic contact network structure of 237 cases in China, distinguishing between three kinds of connections: strangers, weak ties (colleagues, friends), and strong ties (family members, relatives). The number of contacts were unevenly distributed, and 74% of the patients in this population were infected by strong ties, which would indicate clustered infections. The authors suggest that epidemic prevention policies should pay particular attention to ensuring that the various clusters are not interconnected. Consequently, superspreaders are highly relevant for the spread of the coronavirus, as they connect different clusters and not only have the highest chance of becoming infected but also would spread the virus to several other clusters [48].

Due to the current lack of perennial experience and comprehensive data, simulations provide valuable information for possible epidemiological scenarios. The development of mathematical models and theories to enhance our understanding of epidemic processes has long been part of the scientific discourse [49]. Such models simulate possible contacts between infected and susceptible persons to demonstrate the potential spread of a disease using selected factors. Superspreaders are of particular relevance, as they can influence the spread to a great extent. To identify such influential network nodes, epidemic simulations are used to test the performance of different measures [50,51] and to better understand social contagion processes in complex networks [52]. While the already mentioned measures of centrality cannot accurately predict individually relevant nodes, simulations with meta-centralities show more promising results [53]. Nonetheless, how to best quantify and integrate the phenomenon of superspreading into mathematical models remains an open debate [8].

For COVID-19, simulations include the currently available data and insights of the ongoing pandemic. Susceptible-infected (SI) models simulate the spreading process and examine the spreading capability of complex networks [54]. Reich et al. [55] utilized a susceptible-exposed-infected-recovered (SEIR) model, considering among other factors, superspreaders and quarantine policies. They showed, for example, that even with an R0 below 1, exponential spread is possible, since the graph structure significantly influences the growth of the epidemic. Kochańczyk et al. [56] performed stochastic simulations of model dynamics and demonstrated the effect of superspreading events on exponential growth. Through the continuous collection of relevant data and further optimization of mathematical models, our understanding of the viral spread and simulations of interventions can be continuously improved.

### 2.3. Superspreading Events

Special attention should be paid to superspreading events (SSE) where all of the above factors are highly relevant. The integration of epidemiological parameters (including R0, SAR, K), biological and behavioral characteristics (including loud speech, cough-produced aerosols), and social networks (including centrality measures and dynamic network structures) could help to better understand transmission patterns. When several of these factors are present and occur in specific environments, the probability of an outbreak increases accordingly and can lead to an SSE.

At the beginning of COVID-19, several reports of SSEs were published, in which a few infected persons infected a large number of others [57,58]. Furuse et al. [59], for example, analyzed more than 3184 cases of COVID-19 in Japan and identified 61 case clusters. Most of them were associated with heavy breathing in close proximity, like exercising in gymnasiums, singing at karaoke parties, or talking in bars. Further SSEs are associated with explosive growth, as happened in the Diamond Princess cruise ship, which, similar to the Ischgl case, attained inglorious fame. Most damaging, of course, are SSEs that occur in particularly vulnerable environments such as healthcare facilities and retirement or nursing homes [60]. To treat emergency patients in such an environment, rapid clinical intervention is essential [61], taking into account the impact on healthcare professionals, who also face enormous challenges [14,62].

Although several countries have been able to reduce infection rates after the first wave in the sense of “flattening the curve”, SSEs still continue to emerge spontaneously. Currently, for example, in Austria and Germany, cases of individuals who have infected a large number of others in a short period of time in tourist areas, at funerals, or during harvesting on farms are being investigated [63]. These examples illustrate that the exact location and timing of outbreaks in such uncertain situations are difficult to predict; therefore, social systems must be prepared to understand signals and patterns in a short time and intervene immediately [64].

## 3. Managing Superspreading

In the case of COVID-19, a rapid understanding of complex systems is essential but must be integrated with equally rapid action. Althaus et al. [65] showed that in Switzerland carrying out certain interventions only one week later would have significantly increased the number of deaths. Despite intensive global research into possible medical treatments and vaccines against SARS-CoV-2, as long as there are no globally orchestrated solutions, (the preparation of) non-pharmaceutical interventions are currently the only effective countermeasures [66]. According to Fukui and Furukawa [31], even in countries with a low number of infected people, a next wave can suddenly arrive in the future: “to prevent and control of SSEs, speed is essential” [57]. In this section, we will discuss how to rapidly respond to SSEs and highlight prepared improvisation as a key tool to support real-time interventions when routines are not yet established.

### 3.1. Societal Policies and the Role of Information Sharing

In response to COVID-19 superspreaders, various measures have already been taken to limit their impact and slow the rate of virus spread. Governments have introduced measures such as city lockdowns, border closures, identifying cases from other countries, contact tracing, and the use of personal protective equipment [67]. When it comes to the specific behavioral peculiarities that have been associated with SSEs, such as wet pronunciation and the related corona cloud, more emphasis should also be placed on self-awareness, which could have an immediate impact if effectively communicated and thus respected by the population. For example, Japan’s Ministry of Health, Labor, and Welfare announced the simple guiding principle of “avoiding the three Cs” to educate the public: avoid (1) Closed spaces with poor ventilation, (2) Crowded places with many people nearby, and (3) Close-contact settings such as close-range conversations. In environments where these guidelines cannot be implemented, other measures such as continuous monitoring of potentially infectious humans becomes necessary. Depending on the cultural circumstances, public communication and measures must be designed accordingly, which can be anecdotally reflected in how distances between people are depicted (e.g., in Austria the minimum distance was communicated by analogy with the size of baby elephants, in Australia with that of kangaroos). Furthermore, to enhance the compliance of the public, results of recent scientific studies could be communicated in more comprehensible lay terms. For example, Brethouwer et al. [68] simulated small-world networks and showed that reducing long-distance connections significantly reduces the spread of viruses. They proposed a simple principle for communicating the results to the general public: stay nearby or get checked.

To disseminate information [69], the aforementioned network characteristics could be utilized for this purpose. For example, Kathri et al. [70] reported that, compared to previous outbreaks, a higher percentage of the public receives information from YouTube videos (though only half of these videos contain useful information). Like the virus, information can be disseminated worldwide through a number of channels to provide individuals with up-to-date information. Similar to social networks, specific “information superspreaders” (influencers) also have a high impact on the spread of (mis)information [71]. As Yum et al. [72] reported, the messages of Donald Trump and Barack Obama had by far the greatest influence in the social media on the transmission of information on COVID-19. A recently released video, tweeted by Donald Trump, spreading misleading claims about the virus, is a typical example. Although Facebook, YouTube, and Twitter deleted several versions of the video within a few hours, the video went “viral” resulting in more than half a million shares and tens of millions of views [73]. The World Health Organization (WHO) Director-General pointed out in this context that we are fighting not only against a pandemic but also against an infodemic [74]. When dealing with complexity and uncertainties, (over)simplified answers are tempting, potentially even leading to conspiracy theories [75,76]. This illustrates all the more that not only virus superspreaders but also information superspreaders have a great impact.

In sum, educating the public about the importance of favorable factors regarding the emission of droplets and aerosols, (e.g., language, awareness that alcohol consumption can lead to a wetter pronunciation, etc.) and, in connection with this, wearing a mask and keeping distance from other persons, especially indoors, is an important preventive measure. Even if restrictions on bar opening hours or the cancellation of larger events are neither popular nor economically beneficial, such rapid short-term measures are necessary to prevent the further spread of the virus, which could lead to even greater damage. Therefore, clear measures and policies are needed to reduce superspreading events [77]. This is supported by Gasparek et al. [78] whose simulations showed that non-pharmacological interventions that focused on social distancing of particularly vulnerable subgroups reduced the spread [79] and by Rocklöv et al. [64] who found that rapid isolation of patients from the cruise ship Diamond Princess reduced the initially estimated R0 from 14.8 to 1.78.

For the immediate implementation of global measures, it is essential to collect, represent, and interpret data [80]. Web applications such as the COVID-19 Dashboard from the Center for Systems Science and Engineering at Johns Hopkins University provide real-time data for this purpose [81]. In addition to government institutions, citizens have also developed tools, as exemplified by ncov2019.live, which was developed by a 17-year-old high school student in early 2020, collecting and presenting data from the WHO, Center for Disease Control, and other reliable sources, garnering up to 30 million visits from people around the world every day [82]. These examples show that in the case of pandemics, it is essential to ensure not only the dissemination of information and trust in state institutions but also the cooperation of the population, in order to comply with prescribed measures or even proactively provide access to available information.

### 3.2. Fast Response and Prepared Improvisation

Uncertain environments, temporary policies, and (potential) outbreaks affect not only societies but also subsystems such as individual organizations, which also face the challenge of being better prepared for uncertain situations and surprises such as these low-probability, high-impact superspreading events. A short-term, sudden increase in infections requires organizations to develop structures and policies that enable them to respond quickly with limited resources. Dealing with the unpredictable requires long-term preparation and additional procedures and competencies to act in real time [12].

Two attributes characterize these fast-response systems: improvisation and preparedness. Plans are especially useful in stable environments where we have built up appropriate knowledge and can anticipate further developments. By contrast, improvisation refers to deliberate real-time responses to events in the absence of plans [83] and is crucial for crisis management, as plans are never sufficient to deal with critical events with a high degree of uncertainty and ambiguity, such as a pandemic. Organizations must therefore improvise, i.e., act quickly when unpredictable events occur, in order to maintain resilience. Improvisation in this sense is not an arbitrary action but a necessity when time resources are very limited and future events are unforeseeable. This mode of action has already been associated with the COVID-19 pandemic [84], as it allows organizations to act fast in face of a viral outbreak in which planning and execution need to be concurrent [85]. A culture of previous habituation to improvisation as well as empowerment of all involved stakeholders is essential to ensure that people are sufficiently prepared to respond in the absence of a plan. Improvisation involves an element of discomfort and risk which people and systems tend to avoid [86]. In fact, improvised solutions offer no guarantees of success; on the contrary, failure in some contexts can be attributed to the improvisers themselves. This is especially important in the case of the heavily regulated and prescribed sector of healthcare, in which improvisation is often avoided or hidden rather than advertised [87].

Paradoxically, though, improvisation requires preparedness. As discussed by improvisational research, it is not possible to improvise on nothing; rather, improvisation implies previous preparation. Improvisation is prepared spontaneity, based on experience, domain knowledge, and a specific mindset to use freedom within structures [88]. In the scientific community, processes and structures have been adapted to make existing knowledge accessible as quickly as possible, for example with the Public Health Emergency COVID-19 initiative [89]. On a national level, the case of Taiwan is illustrative [90]: the country’s Central Epidemic Command Center produced and implemented a list of 124 measures between January 20 and February 24. Some of these measures were anticipatory (such as the use of artificial intelligence), and some were preemptive (such as the quick cancellation of flights from Wuhan). These were combined with calls for immediate action by the population as a whole. Learning from the previous SARS epidemic was an important part of being prepared and allowed Taiwanese authorities to mobilize the population and to improvise effectively. In the context of the current situation, organizations and countries have an opportunity to learn from current and past experiences. Expecting to return to a world as it was before the pandemic is a risky undertaking. Instead, failures need to be analyzed, and adaptive capacities and competencies further developed in order to create more resilient systems. In addition to contingency plans, real-time preparation for improvised action is a key factor in order to better deal with a next wave or another yet unforeseen threat.

## 4. Conclusions

COVID-19 has shown us how quickly a local event can lead to a global crisis in our interconnected world. Superspreading can be seen as an example of how unpredictable such events can be and reveals that an immediate response is necessary to avoid or at least reduce long-term damages. However, not only do rapid measures have to be implemented, but also complex interconnections must be understood, which usually involve different perspectives [91]. Complex systems of various interacting elements generally elude disciplinary understanding and linear thought patterns. The integration of different perspectives is therefore a prerequisite for the implementation of targeted measures. Arbitrary interventions without systemic understanding are just as unsuitable in emergency situations as the long-term development of strategies without immediate action. In this seemingly paradoxical situation, it is of vital importance to provide all those involved with the available information and resources so that they can act immediately. At the same time, short-term actions enable us to gain further experience and knowledge in order to be better prepared for future events in the long term and thus be able to improvise in an emergency.

The example of COVID-19 superspreaders very clearly shows the inherent complexity of the pandemic. This raises some important questions for future considerations:How can we understand complex adaptive systems better and faster? Understanding systemic interrelationships is necessary to implement effective measures and to consider the resulting consequences for subsequent actions. This requires inter- and transdisciplinary approaches to consider the different viewpoints of science and practice and to integrate both kinds of knowledge for joint decisions.What is the role of scientists and decision-makers in this context? Especially in uncertain situations, the public expects clear answers and decisions. Scientists prefer to gradually improve incomplete knowledge and resist speculation and improvisation. Political decision-makers, on the other hand, would prefer to present the public with clear measures based on solid scientific knowledge. For both sides to work together, they need to better understand their respective roles and preferences.How can we (individuals, organizations, societies) quickly respond to rapidly changing environments? Improvisation is a way of acting and thinking outside of established routines and already existing plans. It is predicated on being prepared, so that in situations of uncertainty requiring immediate decisions, plans and actions must be implemented concurrently and continuously developed on the basis of currently available data and resources, including previous experience.

Understanding and managing global problems requires a networked global world capable of common actions. At the same time, such a world also creates potential risks for future global crises and outbreaks, be they zoonoses, digital viruses, blackouts, or other global threats. COVID-19 has shown us that, starting from a few infected persons, the entire human race can be affected within a few months. In this world, it is no longer enough to look only at one’s own discipline, organization, or nation, but we must develop a common understanding of the existing complexity and implement global measures for global interdependencies.

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
