# Peer review of "Fast Response to Superspreading: Uncertainty and Complexity in the Context of COVID-19"

_ijerph, 2020, doi:10.3390/ijerph17217884_

Round 1

Reviewer 1 Report

Many key citations are missing (e.g. for "A prevailing theory regarding the emergence of viral outbreaks is that human-induced reductions in host species, such as bats, and the deterioration in habitat quality, have made further virus outbreaks even more likely")

Material presented very clearly and in an easy-to-understand way--nice job!

There are two main critiques I have with the manuscript:

  1. It does not seem novel--this would be a great article to present to the public, but very little is new/unknown to scientists
  2. You cite very few mathematical papers--given how important mathematical modeling is to understanding and containing the outbreak, I'm very surprised by this. Before acceptance, I would certainly like to see more mathematical/computational/statistical modeling papers discussed, along with their implications. Along these lines, I would also like to see a discussion of the role that mathematicians/statisticians/computational biologists have in these times.

Author Response

Thank you very much for your detailed and constructive review!

Point 1: Many key citations are missing (e.g. for "A prevailing theory regarding the emergence of viral outbreaks is that human-induced reductions in host species, such as bats, and the deterioration in habitat quality, have made further virus outbreaks even more likely")

Response 1: It is known that the bat-derived virus Pteropine orthoreovirus 3 and MERS coronavirus (MERS-CoV) cause inflammation in human cell lines (Ahn et al., 2019). It is further known that doses of the Ebola virus and MERS-CoV caused only limited pathology in bats, despite high viral tissue titres (Munster et al., 2016). However, questions remain as to whether bats are adapted to tolerate (all) viral infections or simply those they have co-evolved with. Bat longevity and high viral diversity suggest that there are general patterns, but bats are diverse, and whether bats are overrepresented reservoir hosts of zoonotic viral infections is still debated (Olival et al., 2017). Thus, we included these references and rewrote the sentence: “One hypothesis regarding the emergence of viral outbreaks is that human-induced reductions in host species, such as bats, and the deterioration in their habitat quality, may make further virus outbreaks more likely.” (lines 61-63)

Point 2: It does not seem novel--this would be a great article to present to the public, but very little is new/unknown to scientists

Response 2: We agree with you that most of the theories and empirical studies we have described already exist in the scientific literature. Our aim in this publication was not to collect further data or to develop new approaches within the individual disciplines, which are very extensively available in this area. However, we believe that bringing together the various disciplines and arguments in the sense of recombining the existing can provide a new and more holistic perspective of this important phenomenon. In our research we did not find any paper that presented and brought together the knowledge in this form. As a highly interdisciplinary team of authors from the fields of systems sciences, epidemiology, biology, social networks and organizational research, among others, we tried to integrate the different perspectives relevant to this topic in the best possible way. If this undertaking has developed a text that has become understandable for other scientific disciplines or even for the public, we have come one step closer to our goal of an integrative and comprehensible presentation.

Point 3: You cite very few mathematical papers--given how important mathematical modeling is to understand and containing the outbreak, I'm very surprised by this. Before acceptance, I would certainly like to see more mathematical/computational/statistical modeling papers discussed, along with their implications.

Response 3: Thank you for pointing this topic out. We included selected references and a discussion in chapter 2.2 on mathematical/computational/statistical modeling papers (lines 245-266). Because of the continuous collection of empirical data, we are already curious how the mathematical models will improve our understanding of the pandemic and in particular the role of superspreaders. There is, yet, very limited evidence based on modeling specific biological processes. There are, however, empirical studies pointing to inflammation of coronaviruses in human cell lines (see above). One theory, using simple correlation analysis, reported differences in traits related to longevity and infection between bats and humans (Hayman 2019).

Point 4: Along these lines, I would also like to see a discussion of the role that mathematicians/statisticians/computational biologists have in these times.

Response 4: We agree that is an important question that would require a whole critical review, however given the space constraints we cannot really discuss this here in much detail. We included some arguments in chapter 2.2 (lines 245-266) and a reference to one approach in chapter 1, info-metrics, that has produced relevant insights for related infectious diseases, such as TB and SARS (lines 76-78). But this is just one approach among many possibilities.

Reviewer 2 Report

The authors should be congratulated for this very well-written and meticulous review on integrating perspectives from systems sciences, epidemiology, biology, social networks and organizational research in the context of the superspreading phenomenon to understand the complex system and develop suggestions for interventions aimed at rapid responses.

A few suggestions to improve the current version of the manuscript are listed below:

  1. Please spell out the whole abbreviation before using it (COVID-19, SARS-Cov-2, etc)
  2. In page 3 Line 108, the authors should use a reference for the SAR of 35%.
  3. In page 3 Line 147, maybe the authors wanted to use “,” before as well as instead of “;”?
  4. In page 5 Lines 219-220, there is a typo.
  5. Probably adding a couple of figures/tables would make the very interesting content more appealing and easier to grasp to the reader.

Author Response

Thank you very much for your positive review and constructive feedback!

Point 1: Please spell out the whole abbreviation before using it (COVID-19, SARS-Cov-2, etc)

Respond 1: We have written out all abbreviations (COVID-19, SARS-CoV-2, WHO)

Point 2: In page 3 Line 108, the authors should use a reference for the SAR of 35%.

Respond 2: We have included selected references for this estimation (lines 124-125)

Point 3: In page 3 Line 147, maybe the authors wanted to use “,” before as well as instead of “;”?

Respond 3: We corrected this typo

Point 4: In page 5 Lines 219-220, there is a typo.

Respond 4: We corrected the wrong line break

Point 5: Probably adding a couple of figures/tables would make the very interesting content more appealing and easier to grasp to the reader.

Respond 5: Thank you very much for this idea. In an earlier version we did indeed make sketches to visualize our arguments, but we finally decided against using them, because they could only reflect a specific aspect of the arguments or needed concrete parameters that would have to be argued in more detail.

Reviewer 3 Report

A very well written work, bring up an important issue. In the wake of COVId-19, it not only influenced the psychological, sociological, but also medical problems raised in the literature.
This also led to changes in resuscitation guidelines, etc., therefore, I propose to add to the literature some interesting articles that deal with therapeutic problems, e.g.
Prone ventilation of critically ill adults with COVID-19: how to perform CPR in cardiac arrest?
Mędrzycka-Dąbrowska, et al. Critical Care 24 (1), 1-2

Author Response

We are pleased to receive your positive review and agree with you that COVID-19 and its interventions have influenced many different areas.

Point 1: I propose to add to the literature some interesting articles that deal with therapeutic problems, e.g. Prone ventilation of critically ill adults with COVID-19: how to perform CPR in cardiac arrest? Mędrzycka-Dąbrowska, et al. Critical Care 24 (1), 1-2

Response 1: Thank you for the recommendation of the article that we have included (lines 280-282).

Reviewer 4 Report

This work is generally well written and introduces a new perspective on SARS-CoV-2 transmission. Minor points: More abundant reference of previous works concerning healthcare professionals should be commented in the excerpt 163-176 - Wańkowicz, P., et al. Assessment of Mental Health Factors among Health Professionals Depending on Their Contact with COVID-19 Patients. Int. J. Environ. Res. Public Health 2020, 17, 5849.

Author Response

Many thanks for your positive feedback!

Point 1: More abundant reference of previous works concerning healthcare professionals should be commented in the excerpt 163-176 - Wańkowicz, P., et al. Assessment of Mental Health Factors among Health Professionals Depending on Their Contact with COVID-19 Patients. Int. J. Environ. Res. Public Health 2020, 17, 5849.

Response 1: Thank you for the recommendation of the article that we have included, as well as a reference to therapeutic interventions (Mędrzycka-Dąbrowska, et al, 2020). We also included a reference (Steiner et al., 2020), in which we discussed the role of healthcare professionals in detail (lines 280-282).

Reviewer 5 Report

The article by Zenk at al. is well-written and engaging. It explores different aspects of "superspreading". This topic is of interest to a broad audience such as infectious disease specialists, infection control specialists and public health experts.

I do not have any corrections for the authors.

Author Response

Thank you very much for your feedback, which we appreciate very much. Our goal was to work in a highly interdisciplinary team to shed light on an interdisciplinary and socially relevant phenomenon, hoping that it will become more understandable to different readers. Your review encourages us that we may have come a step closer to this goal.

Round 2

Reviewer 1 Report

Thank you for promptly making corrections to many of the points I mentioned in the review--the paper is certainly looking better to me. Your point about synthesizing disciplines and bringing researchers from different areas together is well taken. Though I cannot in good faith rate this paper as having high novelty, I have changed my originality rating from low to average.